

# Variability in Antarctic Surface Climatology Across Regional Climate Models and Reanalysis Datasets

Jeremy Carter[1], Amber Leeson[2], Andrew Orr[3], Christoph Kittel[4], and J.Melchior van Wessem[5]

[1]Department of Mathematics and Statistics, Lancaster University, Lancaster, United Kingdom
[2]Lancaster Environment Centre, Lancaster University, Lancaster, United Kingdom
[3]British Antarctic Survey, High Cross, Madingley Road, Cambridge, UK
[4]Laboratory of Climatology, Department of Geography, SPHERES, University of Liège, Liège, Belgium
[5]Institute for Marine and Atmospheric Research Utrecht, Utrecht University, Utrecht, Netherlands

**Correspondence:** Jeremy Carter (j.carter10@lancaster.ac.uk)

**Abstract.**

Regional climate models (RCMs) and reanalysis datasets provide valuable information for assessing the vulnerability of ice shelves to collapse over Antarctica, which is important for 2100 global sea level rise estimates. Within this context, this paper examines variability in snowfall, near-surface air temperature and melt across products from the MetUM, RACMO and MAR RCMs, as well as the ERA-Interim and ERA5 reanalysis datasets. Seasonal and trend decomposition using Loess (STL) is applied to split the monthly time series at each model grid-cell into trend, seasonal and residual components. Significant, systematic differences between outputs are shown for all variables in the mean and seasonal/monthly standard deviations, occurring at both large and fine spatial scales across Antarctica. It is suggested that differences in the atmospheric dynamics, parametrisation, tuning and surface schemes between models together contribute more significantly to large-scale variability than differences in the driving data, resolution, domain specification, ice sheet mask, digital elevation model and boundary conditions. Despite significant systematic differences, high temporal correlations are found for snowfall and near-surface air temperature across all products at fine spatial scales. For melt, only moderate correlation exists at fine spatial scales between different RCMs and low correlation between RCM and reanalysis outputs. Root mean square deviations (RMSDs) between all outputs in the monthly time series for each variable are shown to be significant at fine spatial scales relative to the magnitude of annual deviations. Correcting for systematic differences results in significant reductions of RMSDs, suggesting the importance of observations and further development of bias-correction techniques.

## 1 Introduction

The largest source of uncertainty in 2100 Sea Level Rise (SLR) projections, for a given Representative Concentration Pathway (RCP), is from the contribution of ice sheets (Kopp et al., 2017). Non-linear instabilities in the Greenland and Antarctic ice sheets give long tails to their SLR probability projections. For example, under RCP 8.5 the median SLR from Antarctica is projected to be of the order of 20 cm, while the 95th percentile is six times higher, at 130 cm (Bamber et al., 2019). The Antarctic continent is fringed by ice shelves, which act like 'ice dams', slowing down the flow of inland ice towards the sea



(Rignot et al., 2004; Scambos et al., 2004). The stability of the ice shelves under a warming climate strongly determines the rate of SLR from Antarctica and it is, in part, the difficulty of modelling their complex physical dynamics, leading to
retreat/collapse, that results in the large uncertainty in estimates of future SLR (Bulthuis et al., 2019).

The primary method of ice shelf retreat is through oceanic basal melting (Pritchard et al., 2012; Paolo et al., 2015), although recent and dramatic collapse events, such as the disintegration of the Larsen B ice shelf in 2002, are linked to anomalous atmospheric conditions through the process of melt-induced hydrofracture (Scambos et al., 2000; van den Broeke, 2005; Bell et al., 2018). Anomalously high near-surface air temperatures (leading to enhanced melt events), as well as low accumulation
(leading to reduced pore space of surface snow), result in greater lateral propagation of melt water into crevasses across the ice shelf, which then deepen due to increased hydrostatic pressure (Kuipers Munneke et al., 2014). This process reduces the structural integrity of the ice shelf and, in addition to fractures created through supraglacial lake filling and drainage, can eventually lead to collapse (Banwell et al., 2013; Kuipers Munneke et al., 2014). Comprehensive spatiotemporal estimates of near-surface air temperature over Antarctica, as well as the accumulation of snowfall and quantity of melt water, are thus
important for SLR predictions and are typically provided by RCMs (van Wessem et al., 2018; Agosta et al., 2019; Mottram et al., 2021).

RCMs are limited-area, physically-based, nested models driven at the boundaries by lower-resolution Global Climate Models (GCMs) or reanalysis datasets. The high-resolution available from RCMs is important for capturing fine-scale climatic processes in regions of complex topography, such as föhn winds that occur over ice shelves on the Antarctic Peninsula (Luck-
man et al., 2014). The region-specific domain enables the set-up and physical schemes of the RCM to be polar optimised (Orr et al., 2021). In addition, further added-value of RCMs is provided through inclusion of region-specific, sophisticated, surface and sub-surface schemes that capture processes such as melt water percolation (Ettema et al., 2010; Datta et al., 2019; Walters et al., 2019). Despite these features, RCMs still exhibit significant systematic errors precluding their direct interpretation in Climate Change Impact Studies (CCIS) (Christensen et al., 2008; Ehret et al., 2012).

The atmospheric model dynamics, surface scheme, parametrisation, driving data, boundary conditions, domain, resolution and orography are all examples of components that contribute to systematic error (Ehret et al., 2012; Giorgi, 2019; Mottram et al., 2021). This paper examines the magnitude and spatial distribution of systematic differences in an ensemble of RCM simulations for Antarctic-wide, 1980-2018 estimates of snowfall, near-surface air temperature and melt water. The relative contribution from different components of the simulations, such as the atmospheric model physics, are discussed. Comparisons
of Antarctic-wide RCM simulations of recent-historic surface climatology are present in the literature (Mottram et al., 2021; van Wessem et al., 2018; Agosta et al., 2019), although the focus is predominantly on Surface Mass Balance (SMB). Surface melt flux, when integrated over the Antarctic ice sheet, only represents a small fraction of the total SMB, which is determined predominantly by the flux of snowfall (Lenaerts et al., 2012b; Agosta et al., 2019). This paper provides the first inter-comparison of recent-historic Antarctic-wide RCM simulations framed within the context of ice shelf instability and collapse events, giving
specific focus to variability in near-surface air temperature, snowfall and melt water.

Six Antarctic-wide RCM simulations are compared, two from each of the Met Office Unified Model version 11.1 (Me-tUMv11.1), the Modèle Atmosphérique Régional version 3.10 (MARv3.10) and the Regional Atmospheric Climate Model



| RCM/Reanalysis Dataset | Domain | Driving Data | H.Resolution [km] | Label |
|---|---|---|---|---|
| ERA-Interim | Global | - | 79 | ERAI |
| ERA5 | Global | - | 31 | ERA5 |
| MetUMv11.1 | Antarctica | ERA-Interim | 12 | MetUM(011) |
| MetUMv11.1 | Antarctica | ERA-Interim | 49 | MetUM(044) |
| MARv3.10 | Antarctica | ERA-Interim | 35 | MAR(ERAI) |
| MARv3.10 | Antarctica | ERA5 | 35 | MAR(ERA5) |
| RACMOv2.3p2 | Antarctica | ERA-Interim | 27 | RACMO(ERAI) |
| RACMOv2.3p2 | Antarctica | ERA5 | 27 | RACMO(ERA5) |

**Table 1.** The two reanalysis datasets and six RCM simulation outputs compared in the paper. The label with which each simulation is referred to in the paper is given.

version 2.3p2 (RACMOv2.3p2). Comparisons are also made to the reanalysis driving data of ERA-Interim and ERA5. The resulting eight Antarctic-wide datasets analysed in this paper are given in Table 1. MARv3.10 and RACMOv2.3p2 are both

hydrostatic models specifically developed for use over polar regions and their output from Antarctic-wide simulations have been rigorously compared to one-another and against observations (Lenaerts et al., 2012b; van Wessem et al., 2018; Agosta et al., 2019). The MetUMv11.1 is not specifically developed with a focus on the polar regions, although it is a non-hydrostatic model meaning it can be run at and simulate atmospheric circulation features at sub-kilometer resolutions (Orr et al., 2021), whereas MAR and RACMO are limited to maximum resolutions of 5-10 km horizontal grid spacing (van Wessem et al., 2016;

Datta et al., 2019). Further detail on key differences in the model specifications for the simulations analysed in this paper, such as surface schemes used, is presented in section 2.

Historic, evaluation simulations are chosen to remove dependency on emission scenarios, which have been shown to introduce divergent trajectories of variables such as melt (Trusel et al., 2015; Gilbert and Kittel, 2021; Kittel et al., 2021). Comparisons to observations are not included due to the sparse nature of observations available over Antarctica. Papers in-

cluding observations typically require comparisons to be made across elevation bins (Mottram et al., 2021; van Wessem et al., 2018; Agosta et al., 2019). In this paper comparisons are made at a 12 km grid-cell level and it is shown that variability between the simulations has greater dependency on the (latitude, longitude) location than elevation. In addition, to study the temporal dependence of variability, time series decomposition is applied, separating the signal at each location into an annual, seasonal and residual component. These components are driven by different physical processes and the previous inter-comparison pa-

pers cited have not focused on examining the nature of variability at different temporal scales. Finally, despite the primary motivation for this paper focusing on surface climatology over ice shelves, the analysis is extended to the whole Antarctic ice sheet and surrounding Southern Ocean. This is done to aid discussion, as surface climatology over the ice shelves is influenced by the behaviour of the models over the rest of the domain, and extending the analysis provides insights useful for studies not only focused on ice shelves, thus increasing the scope of the work.





## 2  Reanalysis Datasets and RCMs Specifications

The ensemble of Antarctic-wide RCM simulations examined in this paper are part of the Coordinated Regional Climate Down-scaling Experiment (CORDEX: https://cordex.org/), which is a global project that provides coordinated sets of RCM simulations worldwide. The model specifications for each of the RCM simulations in the chosen ensemble, as well as for the ERA-Interim and ERA5 reanalysis products, are detailed here. There are significant differences, with some of the key aspects being: different atmospheric dynamics components between all models; different surface schemes - of particular note is the 'zero-layer' scheme used in the MetUM simulations, which has been identified as a major deficiency in simulations, compared with the multi-layer schemes included in MAR and RACMO, due to such impacts as that on heat transfer and not representing the insulating properties of the column of snow (Slater et al., 2017; Walters et al., 2019); differences in the vertical and horizontal resolutions between all models, with particular interest on the performance of the high-resolution 12 km MetUM against the low-resolution 49 km simulation; differences in the driving data between the two RACMO and two MAR simulations, that are otherwise identical; differences in the Digital Elevation Models (DEMs) and masks used by each model, with MAR and RACMO using comparatively similar DEMs, while the MetUM uses a DEM similar to that of ERA5 but shown to have large-scale differences relative the that used by MAR and RACMO of over 100 m, particularly over areas close to the perimeter of the ice sheet (Fig. C1).

### 2.1  ERA-Interim and ERA5

ERA-Interim, produced by ECMWF, is a global reanalysis dataset, spanning 1979-2019 with 6-hourly temporal resolution, approximately uniform horizontal resolution of 79 km spacing and 60 vertical levels up to 10 Pa (Dee et al., 2011). The ECMWF uses a four-dimensional variational data assimilation technique (4D-Var). Era-Interim was world leading and is included as the specified driving data in the base criteria for the CORDEX simulations but has since been superseded by ERA5, also produced by ECMWF (Hersbach et al., 2020), with a number of ERA5 driven simulations also included in the Antarctic-CORDEX ensemble of RCM outputs. The ERA5 reanalysis dataset uses the updated Cycle 41r2 version of the Integrated Forecast System (IFS) numerical weather prediction (NWP) model, with significant developments to model physics and assimilation methods (Hersbach et al., 2020). It spans 1950-Present with an enhanced single hourly temporal resolution, horizontal resolution of 31 km and 139 vertical levels up to 1 Pa. In addition, ERA5 has uncertainty estimates derived from an ensemble of 10 data assimilations performed at a 3 hourly temporal resolution and horizontal resolution of 63 km. The elevation used by ERA-Interim comes from interpolating the GTOPO30 elevation product (ECMWF, 2009), whereas for ERA5 surface elevation is derived from interpolation of a combination of the SRTM30 elevation product along with other surface elevation datasets (ECMWF, 2016). The coupled surface schemes used for ERA-Interim and ERA5 are the Tiled ECMWF Scheme for Surface Exchanges over Land (TESSEL) and updated HTESSEL schemes respectively, both use a single tile to represent snow, while one of the major differences is that HTESSEL allows surface runoff (Balsamo et al., 2009).



## 2.2 MAR

MAR is a hydrostatic RCM, specifically developed for the polar areas (Fettweis et al., 2013). The Antarctic-wide simulations analysed in this paper have a spatial horizontal resolution of 35 km with a vertical resolution of 24 atmospheric levels. Specific details of the atmospheric component of MAR can be found in Gallée and Schayes (1994); Gallée (1995). The atmospheric model is fully coupled to the 1-D SISVAT (Soil Ice Snow Vegetation Atmosphere Transfer) surface scheme (Fettweis et al., 2013, 2017), which uses the Crocus multi-layer surface snow model (Brun et al., 1992) that contains subroutines for processes such as: snow metamorphism and meltwater runoff, retention, refreezing and percolation. SISVAT does not include a full radiative transfer scheme in snow/ice and surface albedo is parameterised as a function of snow grain properties (Tedesco et al., 2016). The relaxation technique is used to apply LBCs from the driving data every 6 hours and spectral nudging is used to constrain the large-scale behaviour in the upper atmosphere. The two Antarctic-wide MAR simulations studied in this paper are identical apart from differing driving data from ERA-Interim and ERA5 respectively. The orography used in the simulations is from BEDMAP2 (Fretwell et al., 2013). For further detail on MAR and the specific version used to generate the output examined in this paper (MARv3.10) the reader is referred to Agosta et al. (2019) and Mottram et al. (2021).

## 2.3 RACMO

RACMO is a hydrostatic RCM with a polar version developed to represent the climate specifically over ice sheets (Van Meijgaard et al., 2008). The RCM uses the dynamical core from HIRHAM (High Resolution Limited Area Model) (Undén et al., 2002) and the physics package CY33r1 version of the Integrated Forecast System (IFS) NWP model from ECMWF. The Antarctic-wide simulations analysed in this paper have a spatial horizontal resolution of 27 km with a vertical resolution of 40 atmospheric levels. The simulations include a multi-layer snow scheme that simulates hydrological processes such as melt, percolation, refreezing and runoff as well as firn densification (Ettema et al., 2010). In addition, a drifting snow scheme simulates movement of snow from surface winds across the ice sheet (Lenaerts et al., 2010, 2012a). A snow albedo scheme is implemented, which uses snow grain size as a prognostic variable as well as cloud optical thickness and solar zenith angle to estimate albedo (Munneke et al., 2011). The relaxation technique is used to apply LBCs from the driving data every 6 hours for the RACMO simulation driven by ERA-Interim and every 3 hours for the simulation driven by ERA5 and spectral nudging is used to constrain the large-scale behaviour in the upper atmosphere. The two simulations studied are identical apart from differing driving data from ERA-Interim and ERA5 respectively. The orography used in the simulations is the same as from Bamber et al. (2009). For further detail on RACMO and the specific version used to generate the output examined in this paper (RACMOv2.3p2) the reader is referred to van Wessem et al. (2018) and Mottram et al. (2021).

## 2.4 MetUM

The MetUM is a non-hydrostatic climate model, not specifically developed or optimised for use over the polar regions but adapted in these simulations for use over Antarctica (Orr et al., 2021). The Regional Atmosphere physics configuration for mid-latitudes (RA1M) is used (Bush et al., 2020), which is identified as the most suitable configuration available for simulating near-

 

surface climatology over Antarctica (Gilbert et al., 2020, 2021). The Joint UK Land Environment Simulator (JULES) (Walters et al., 2019) is used with the option of a comparatively simple zero-layer snow/soil composite scheme that does not capture

processes such as refreezing of melt water (Best et al., 2011). The two Antarctic-wide MetUM simulations analysed in this paper are identical apart from their spatial horizontal resolutions of 12 km and 49 km respectively, both have a common vertical resolution of 70 atmospheric levels. These limited-area, regional simulations are nested inside the global model configuration of the MetUM, which is itself forced using ERA-Interim reanalysis data and follows a 12 hour re-initialisation procedure that constrains the large-scale circulation in the interior of the domain and prevents it from drifting too far from the driving data

(Gilbert et al., 2021). The global MetUM model runs for 24 hour periods, with a re-initialisation happening throughout the domain every 12 hours and boundary conditions for the nested run saved each hour. The first 12 hours of each 24 hour run is discarded as spin-up, while the second 12 hours of each run is kept as output and stitched together with following runs. The orography used in the simulations is the MetUM standard GLOBE 1 km dataset (Elvidge et al., 2019).

## 3    Comparison Method

The RCM simulations examined in this paper all use an equatorial rotated coordinate system, where a quasi-uniform horizontal-resolution grid is defined over the region by first specifying the grid over the equator with constant latitude and longitude spacing between each grid-cell and then applying a rotation that takes the domain over the region of interest, for example Antarctica. Direct comparisons between the model output are made by regridding onto a common grid, with a common domain and spatiotemporal coordinates. Cubic precision Clough-Tocher interpolation (Mann, 1999) is performed on the unrotated 'grid

latitude' and 'grid longitude' coordinates, which are assumed approximately euclidean, to regrid all model outputs onto the MetUM(011) resolution grid. This grid is chosen as it is the highest resolution grid of the simulations examined, meaning no information is lost as part of the regridding. The domain is filtered to only include the regions common across the model outputs, see Fig. 1. The time series examined is filtered to the common 1981-2018 period and 3/6 hourly outputs are aggregated to monthly averages, which captures the dominant annual and seasonal dependency in the variability. For surface air temperature,

filtering to only the common timestamps across the models is first applied and then the average temperature over each month computed. The common timestamps are limited by ERA-Interim to 00 h, 06 h, 12 h and 18 h. This is not required for snowfall or melt, which are defined as fluxes in the model output.

To study annual, seasonal and monthly variability separately, Seasonal and Trend decomposition using Loess (STL) (Cleveland et al., 1990) is applied to the time series of each variable at each grid-cell. This results in individual trend (T), seasonal

(S) and residual (R) components. The decomposition is additive, meaning for each data point $\nu=1$ to N, the components are summed to give the original time series (Y) (eq.[ 1]). The trend component represents the low-frequency/long-timescale pattern of the time series, after filtering out medium and high-frequency signals including the seasonal component, which captures



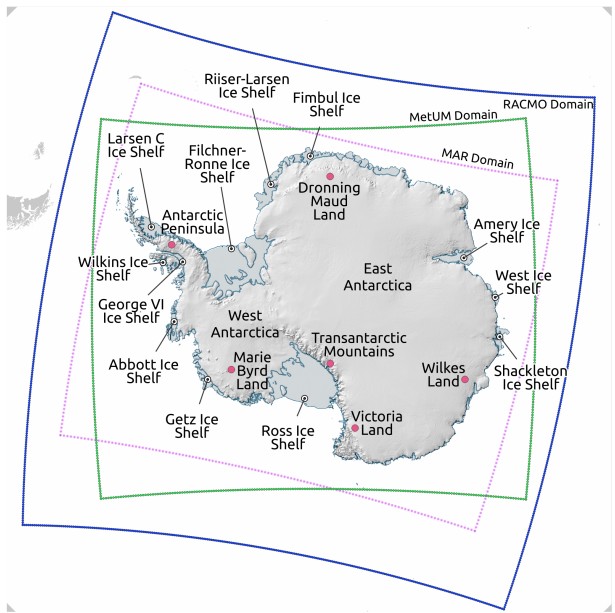

**Figure 1.** Map of Antarctica with some of the main regions and ice shelves labelled, made using the Quantarctica mapping environment (Matsuoka et al., 2021). The RCM simulation domains for the MetUM (green), RACMO (blue) and MAR (purple) are shown. A 1 km resolution hill-shade has been applied from BEDMAP2 (Fretwell et al., 2013).

periodic patterns, and the residual component that explains fluctuations not caused by the long-scale trend or periodicity in the time series.

$$Y_\nu = T_\nu + S_\nu + R_\nu \tag{1}$$

Basic time series decomposition involves first approximating the trend component by applying a polynomial fit through the data. Subtracting this component gives the de-trended data that is then split into seasonal sub-series (e.g. January, February, ...) and an average of each sub-series gives the seasonal component of the data. Subtracting both the trend and seasonal components then gives the residual component of the series. STL is a more sophisticated procedure that allows options such as robust fitting

(where the influence of outliers is limited) and also a time-varying seasonal component. The algorithm is iterative and involves two loops: the outer loop reduces the influence of outliers by assigning weights based on the magnitude of the remainder term; the inner loop involves estimation of the trend and seasonal components through iterative feedback (Cleveland et al., 1990).

The seasonal component is allowed to vary smoothly over the time series, which is done by applying a LOESS (LOcal RegrESSion) smoothing to the monthly sub-series with window length $n_s$. As $n_s \to \infty$ the LOESS smoothing becomes equivalent

to simply taking the average over the sub-series. The value of $n_s$ is recommended to be greater than 7 (Cleveland et al., 1990). As the value increases, the seasonal component approaches a constant periodic state. In this work 13 is used as this allows





potential decadal oscillations in the climate to be captured in the seasonal component, such as the Pacific Decadal Oscillation (PDO).

The trend component is estimated using LOESS with a window of default size ($n_t$) given by the smallest odd integer greater
than the value in eq.[2], which for a period ($n_p$) of 12 months and seasonal smoother ($n_s$) of 13 gives $n_t = 21$. For intuition, this means the seasonal component can be thought of as a 12 month periodic signal that is allowed to change gradually over a 13 year period, while the trend component can be thought of as similar to the result of taking a weighted moving average of the deseasonalized time series over a 21 month period. The residual component is then the remaining signal not described by either the smoothly varying seasonal cycle or the long-timescale trend. An example of applying STL decomposition to the
time series of snowfall, surface temperature and melt for a grid-cell on the Larsen C ice shelf is available in section A of the appendix.

$$n_t \geq \frac{1.5 n_p}{1 - 1.5 n_s^{-1}} \tag{2}$$

In this paper temporal variability between the ensemble of Antarctic-wide datasets is assessed in several ways, including: calculating the Pearson linear correlation coefficient between the outputs for each component of the time series and each
variable of interest; quantifying differences in the mean of the time series as well as in the standard deviation of the seasonal and residual components; and calculating the RMSD between the outputs for each variable of interest. Each metric is calculated for every grid-cell in the domain, with Antarctic-wide plots showing spatial patterns. Differences in the monthly mean and standard deviation of the components are calculated over the 37 year 1981-2018 period. For snowfall and melt, differences at each grid-cell are expressed as a proportion of the respective inter-annual deviations, providing some measure of the relative
significance of differences at each location. The impact of systematic differences on estimates of ice shelf stability depend not only on absolute magnitudes but also on the relative magnitude against a baseline variance, as well as how this influences the ratio in magnitudes between snowfall and melt. The inter-annual, baseline deviation at each grid-cell is approximated as the ensemble average standard deviation in the trend component of the time series. For near-surface air temperature differences are not expressed as a proportion and instead simply in degrees Kelvin.

## 210  4  Results

Variability in the ensemble of Antarctic-wide outputs (Table 1) for the monthly time series of snowfall, near-surface air temperature and melt are quantified across the domain through the evaluation of metrics including the correlation between the outputs, systematic differences in the mean and seasonal/residual standard deviations as well as the RMSDs between outputs. These metrics, for variability in the time series, are evaluated at each grid-cell and the main results shown in sections 4.1, 4.2
and 4.3. Spatial maps are used to show large and small scale patterns in the metrics across the domain. Discussion around the results, including features of variability and the relative importance of contributing factors, is given in section 5.



## 4.1 Correlation

Results are presented for the correlation in the deseasonalized and detrended, residual component of the time series between each of the 28 unique model output pairs. The correlation is computed at every grid-cell and for melt, grid-cells where the ensemble 40-year average monthly melt is less than 1 millimeter water equivalent per month ($mmWEqm^{-1}$) are masked as these regions only experience sporadic and insignificant magnitude melt events, essentially equating to numerical noise in the simulations. The average grid-cell correlation across the entire ice sheet is then taken and the results given in Fig. 2. High correlation is shown for snowfall (>0.80) and near-surface air temperature (>0.90) across all model pairs, while results for melt show a significant divide between the reanalysis datasets and the RCMs. The correlation for melt between just the RCMs is moderate to high (>0.55) across all pairs, while for the reanalysis datasets the correlation is low (<0.35) for comparisons to all other models, including between ERA-Interim and ERA5. Another key feature includes the comparatively high correlation shown in every variable between simulations of the same RCM but differing resolution/driving data (MetUM(044)-MetUM(011), MAR(ERAI)-MAR(ERA5) and RACMO(ERAI)-RACMO(ERA5)).

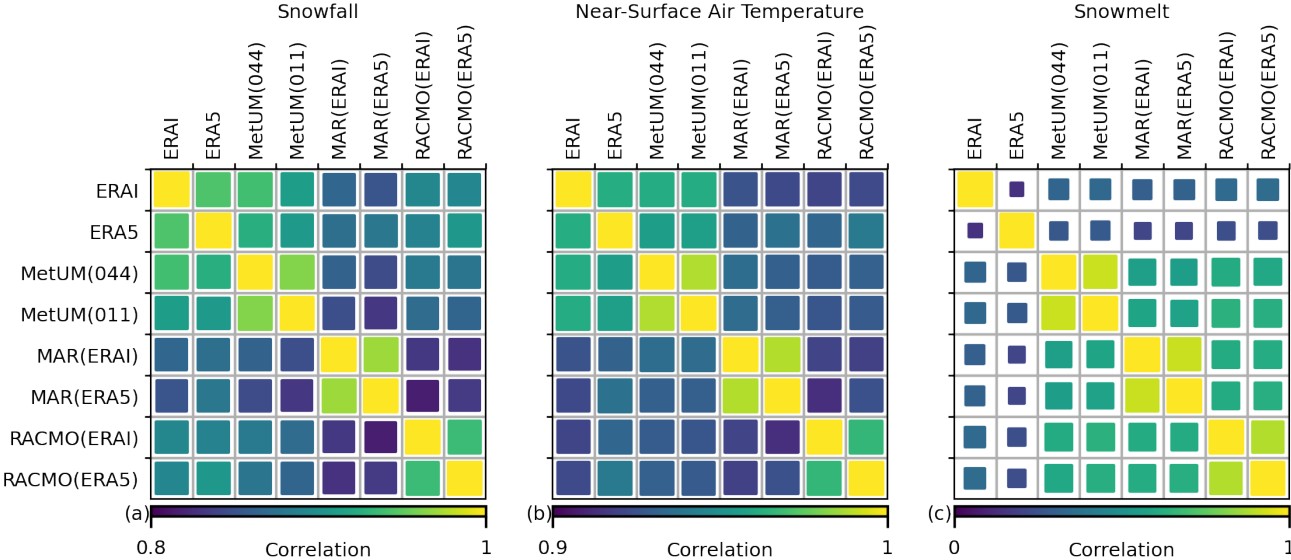

**Figure 2.** The correlation for snowfall (a), near-surface air temperature (b) and melt (c) between models averaged over the ice-sheet. The colour scale is different for each plot and relates to the value of correlation. The size of each square also relates to the magnitude of correlation and the scale for this is constant across the figures, going from 0 to 1.

A spatial map of the median correlation in the residual component across the 28 unique model output pairs is plot in Fig. 3. An ice sheet-only mask is applied for melt using the high resolution shapefile from Depoorter et al. (2013), which is found to remove the most prominent edge effects caused by comparing high- and low-resolution models for a variable that is dependent on the sea/ice categorisation of the grid-cell. In addition, grid-cells where the ensemble 40-year average melt is less than 1





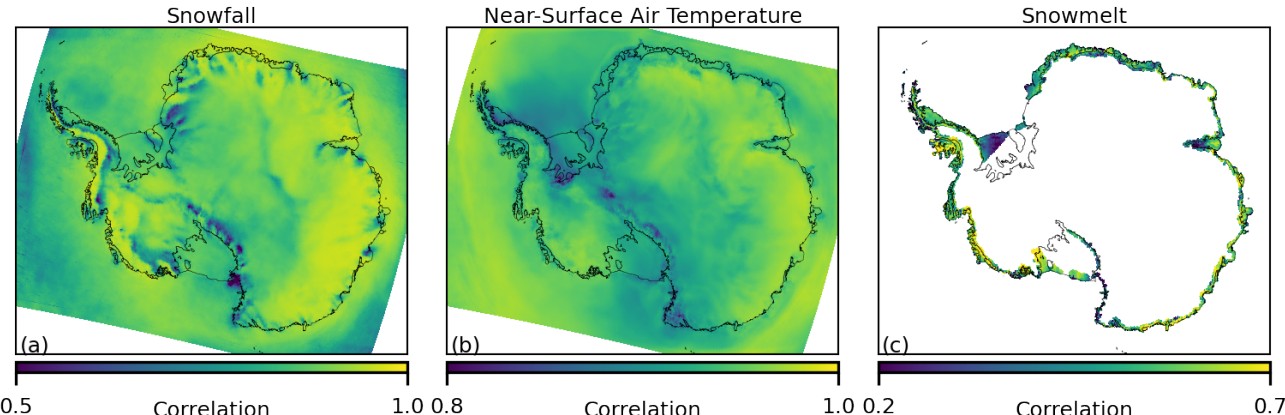

**Figure 3.** The median correlation by grid-cell in the residual component of the monthly time series between the 28 unique model pairs for snowfall (a), near-surface air temperature (b) and melt (c).

millimeter water equivalent per month ($mmWEqm^{-1}$) are again masked. In Fig. 3 the median correlation for near-surface air temperature is shown to be high (>0.8) across the ice sheet, while for snowfall the correlation remains high again across the majority of the ice sheet but is moderate to low over regions such as the trans-antarctic mountains, where the topography varies sharply. For melt, the correlation is moderate over the majority of ice shelves, although is noticeably low over: the Ronne ice shelf; the ice shelves bounding Victoria Land; and the interior of the Amery ice shelf.

### 4.2 Mean and Standard Deviation: Magnitude and Spatial Pattern of Differences

The 1981-2018 mean and standard deviation for each component of the monthly time series of the ice sheet total snowfall, average near-surface air temperature and total melt are displayed in Table 2. Results show that even aggregated across the entire ice sheet significant systematic differences exist between the outputs for each variable. For example, the magnitude of differences in the mean across the ensemble are comparable in magnitude to the average trend standard deviation, which represents inter-annual variations. One particularly striking feature is the contrast between the low monthly melt of ERA5 (1.1 GT/month) compared to the high monthly melt of ERA-Interim (15.5 GT/month) and all RCMs (9.1-14.2 GT/month). It is noted that the relative magnitudes of standard deviations in each component of the time series and that for temperature and melt the seasonal deviation is dominant, while for snowfall both the seasonal and residual deviations have similar magnitudes. Another feature, is that systematic differences are comparatively low between simulations of the same RCM but differing resolution/driving data (MetUM(044)-MetUM(011), MAR(ERAI)-MAR(ERA5) and RACMO(ERAI)-RACMO(ERA5)) when compared with differences present between the different RCMs.

To understand how systematic differences vary spatially the 1981-2018 mean and seasonal/residual standard deviations for the monthly time series of each variable are also computed at a 12 km grid-cell level. Since it is found that systematic differ-





| Snowfall / GT | ERAI | ERA5 | MetUM(044) | MetUM(011) | MAR(ERAI) | MAR(ERA5) | RACMO(ERAI) | RACMO(ERA5) | Average |
|---|---|---|---|---|---|---|---|---|---|
| Monthly Mean | 179.3 | 225.8 | 212.4 | 222.8 | 234.9 | 235.6 | 229.6 | 231.3 | 221.5 |
| Trend St.D. | 7.1 | 9.8 | 9.6 | 9.8 | 8.7 | 8.8 | 8.6 | 9.1 | 8.9 |
| Seasonal St.D. | 25.9 | 34.1 | 26.8 | 25.8 | 38.7 | 38.8 | 30.1 | 31.0 | 31.4 |
| Residual St.D. | 21.8 | 28.3 | 28.3 | 28.9 | 26.3 | 26.3 | 28.0 | 28.2 | 27.0 |
| **Temperature / K** | ERAI | ERA5 | MetUM(044) | MetUM(011) | MAR(ERAI) | MAR(ERA5) | RACMO(ERAI) | RACMO(ERA5) | Average |
| Monthly Mean | -32.6 | -33.3 | -34.2 | -33.9 | -32.2 | -32.2 | -34.0 | -33.8 | -33.3 |
| Trend St.D. | 0.4 | 0.4 | 0.4 | 0.4 | 0.4 | 0.4 | 0.4 | 0.5 | 0.4 |
| Seasonal St.D. | 9.0 | 7.7 | 9.3 | 9.2 | 8.7 | 8.6 | 8.8 | 8.7 | 8.7 |
| Residual St.D. | 1.1 | 1.1 | 1.0 | 1.0 | 1.0 | 1.0 | 1.2 | 1.2 | 1.1 |
| **Melt / GT** | ERAI | ERA5 | MetUM(044) | MetUM(011) | MAR(ERAI) | MAR(ERA5) | RACMO(ERAI) | RACMO(ERA5) | Average |
| Monthly Mean | 15.5 | 1.1 | 13.2 | 14.2 | 11.9 | 12.1 | 9.3 | 9.1 | 10.8 |
| Trend St.D. | 2.4 | 0.4 | 3.0 | 3.1 | 3.3 | 3.1 | 2.9 | 2.7 | 2.6 |
| Seasonal St.D. | 29.3 | 2.0 | 25.8 | 27.0 | 23.1 | 23.4 | 18.5 | 18.2 | 20.9 |
| Residual St.D. | 5.3 | 0.8 | 6.8 | 6.9 | 7.0 | 6.8 | 7.1 | 6.7 | 5.9 |

**Table 2.** After aggregating across the ice sheet, the mean and standard deviation for each component of the monthly time series for total snowfall, average near-surface air temperature and total melt are given. Values for snowfall and melt are expressed in units of gigatonnes while values for temperature are expressed in Kelvin.

ences in the mean and standard deviations are most pronounced between different models in the ensemble, results presented in Fig. 4, 5 and 6 are filtered to only include: ERA5; MetUM(011); MAR(ERA5); and RACMO(ERA5). Results showing direct comparisons between same/similar model pairs are plot in Fig. B1, B2 and B3 in the appendix and include differ-
ences in the mean and standard deviations between: ERA-Interim and ERA5; MetUM(044) and MetUM(011); MAR(ERAI) and MAR(ERA5); RACMO(ERAI) and RACMO(ERA5). Differences in the standard deviation of the trend component are excluded from grid-cell level results as it is shown in Table 2 that the relative magnitude against standard deviations in the seasonal and residual components is low. For snowfall and melt, differences at each grid-cell are expressed as a proportion of the respective inter-annual deviations, approximated by the ensemble average standard deviation in the trend component. This is
done so results presented in spatial maps show the relative significance of systematic differences and are not simply dominated by the sites with the highest magnitude snowfall/melt. Given that for near-surface air temperature, it is not clear whether higher magnitude differences in the mean and standard deviation are expected for sites with greater average temperature, then instead of expressing the results as a proportion they are simply expressed in degrees kelvin.

In Fig. 4, it can be seen that for snowfall there exists significant systematic differences over both the ocean and ice sheet,
particularly in the mean of the time series (Fig. 4a,d,g,j), for each model relative to the ensemble average. A specific example is the strong negative difference in the mean snowfall over the ocean and strong positive difference over the majority of the ice sheet shown by MAR (Fig. 4g). In general, the +ve/-ve sign of the differences in the mean and standard deviations for snowfall over the interior of the ice sheet, over the trans-antarctic mountains and over the oceans show a relatively large spatial correlation length scale. In contrast, near the periphery of the ice sheet, the sign of the differences exhibit a smaller correlation



length scale. It can be seen that for snowfall the difference present in the mean of the time series is highly correlated with the difference in the standard deviation of the seasonal and residual components (e.g. Fig. 4g,h,i). Exceptions to this include for example the difference in snowfall from the MetUM(011) relative to the ensemble in the interior of East Antarctica, where despite having a lower mean snowfall the standard deviation in the seasonal component is greater than the average of the ensemble (Fig. 4d and 4e).

As with snowfall, there exists significant differences over both the ocean and land for near-surface air temperature between the models, again particularly in the mean of the time series (Fig. 5). For example, MAR shows a significant positive difference in the mean of the time series over the majority of the ice sheet (Fig. 5g) and a significant negative difference over the majority of the surrounding ocean. The spatial patterns of differences in near-surface air temperature differ in shape compared to those present for snowfall. In particular, near the edge of the ice sheet there are less positive-to-negative fluctuations with changing

longitude and instead the patterns are more parallel to the coastline (Fig. 5d,g). Unlike for snowfall, there is also a weak, negative correlation between the mean temperature difference and the seasonal standard deviation difference. This suggests that a colder mean temperature may largely be the result of similar summer temperatures with more severe winter temperatures, as can be observed in the case of the example grid-cell in Fig. A1e in the appendix.

A land-only mask has been applied for melt in Fig. 6 as well as a filter masking any grid-cells where the ensemble mean

average monthly melt is less than $1 \, mmWEqm^{-1}$. This limits discussion of the patterns in differences of the mean and standard deviations to the peripheral areas, which are predominantly ice shelves. The magnitude of the differences present is, as for snowfall, significant relative to the inter-annual variability of melt at each grid-cell. Unlike for snowfall and near-surface temperature, the relative strength of differences in the mean of the time series is closer in magnitude to the differences in the standard deviation of the seasonal and residual components. As with near-surface temperature and snowfall there are both short

and long spatial length scale patterns. An example of a relatively localised spatial pattern is that of the strong positive difference shown by MAR over the interior of the Amery ice shelf in the mean of the time series, as well as the standard deviation of the seasonal and residual components. An example of a large-scale pattern is that ERA5 shows a considerable negative difference in the mean and standard deviations of melt over the majority of ice shelves.



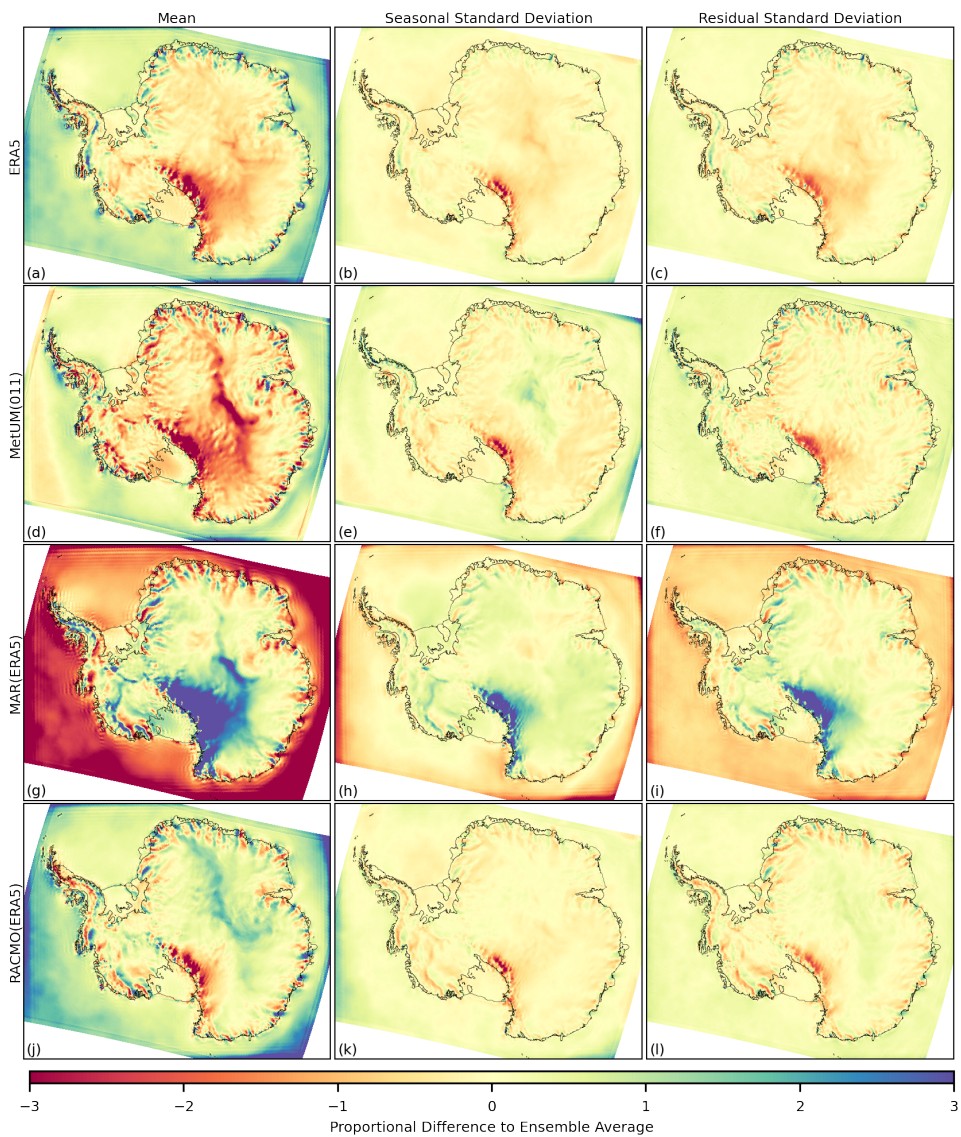

**Figure 4.** The difference to the ensemble average for the 1981-2018 time series of snowfall, in the mean (a,d,g,j), the standard deviation of the seasonal component (b,e,h,k) and the standard deviation of the residual component (c,f,i,l). The ensemble includes: ERA5 (a,b,c); MetUM(011) (d,e,f); MAR(ERA5) (g,h,i); and RACMO(ERA5) (j,k,l). Differences at each grid cell are expressed as a proportion of average inter-annual variation and so do not have units.




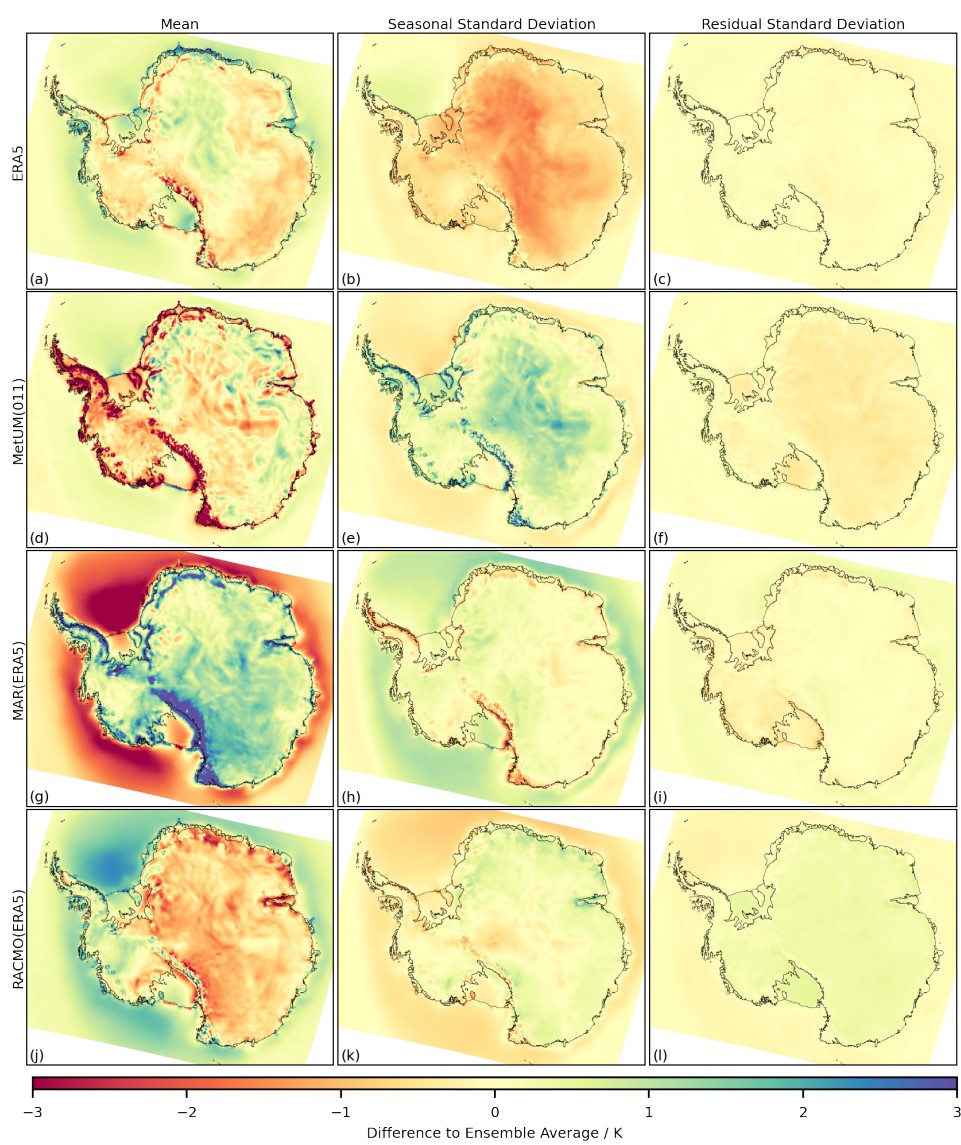

**Figure 5.** The difference to the ensemble average for the 1981-2018 time series of near-surface air temperature, in the mean (a,d,g,j), the standard deviation of the seasonal component (b,e,h,k) and the standard deviation of the residual component (c,f,i,l). The ensemble includes: ERA5 (a,b,c); MetUM(011) (d,e,f); MAR(ERA5) (g,h,i); and RACMO(ERA5) (j,k,l).





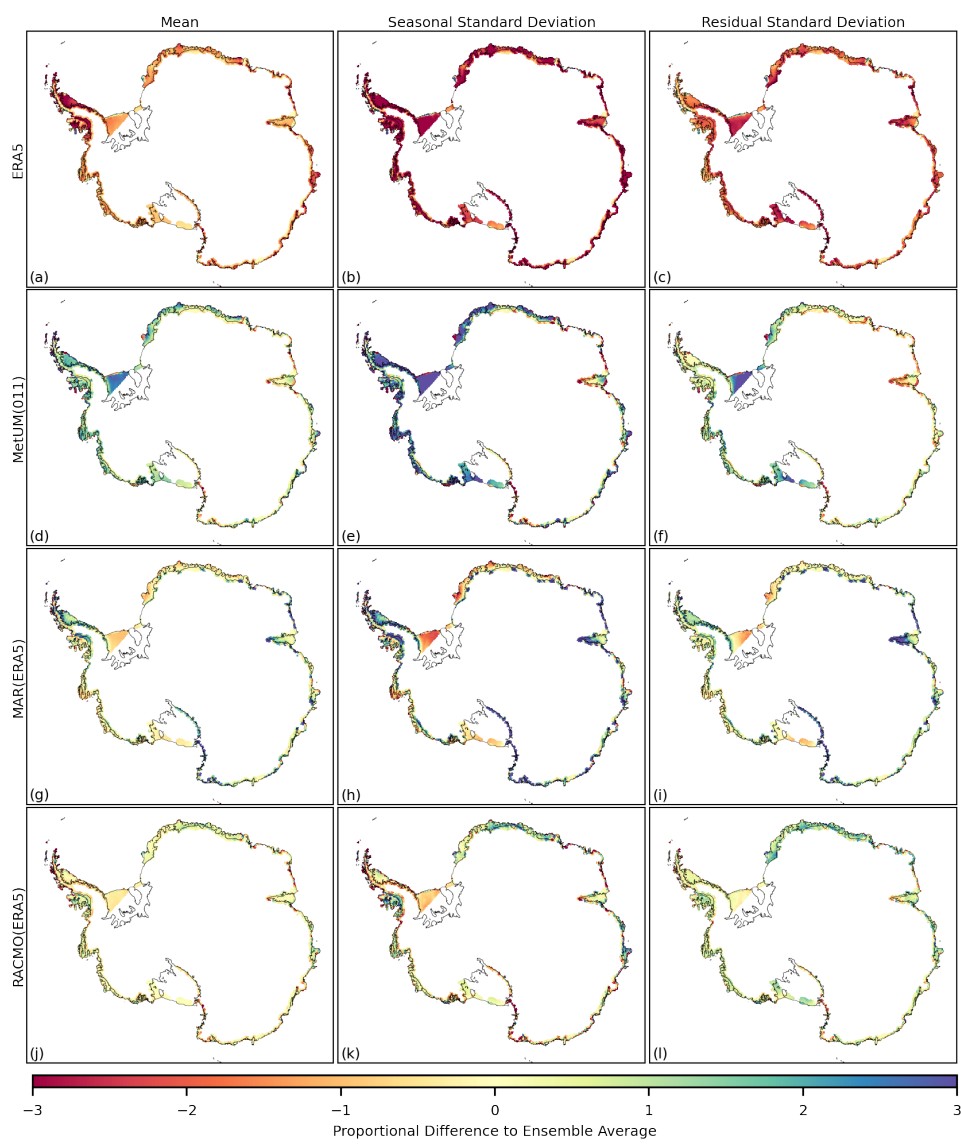

**Figure 6.** The difference to the ensemble average for the 1981-2018 time series of melt, in the mean (a,d,g,j), the standard deviation of the seasonal component (b,e,h,k) and the standard deviation of the residual component (c,f,i,l). The ensemble includes: ERA5 (a,b,c); Me-tUM(011) (d,e,f); MAR(ERA5) (g,h,i); and RACMO(ERA5) (j,k,l). Differences at each grid cell are expressed as a proportion of average inter-annual variation and so do not have units.




## 4.3 RMSD

The RMSD and the percentage reduction in RMSD after adjusting for differences in both the mean and seasonal/residual standard deviations of the monthly time series is evaluated at each grid-cell for each of the 28 unique output pairs of the ensemble. For snowfall and melt, the metric is scaled at each grid-cell by the inter-annual standard deviation, giving the relative significance of the RMSD value and described here as the proportional RMSD value. After masking to the ice sheet only and applying an additional mask for melt, where only grid-cells with greater than $1\ 1mmWEq$ of melt per month are

included, the average is taken across the ice sheet and results given in Fig. 7. The percentage change in the RMSD/proportional RMSD after adjusting the mean and seasonal/residual standard deviations of all outputs to that of the ensemble average is also given.

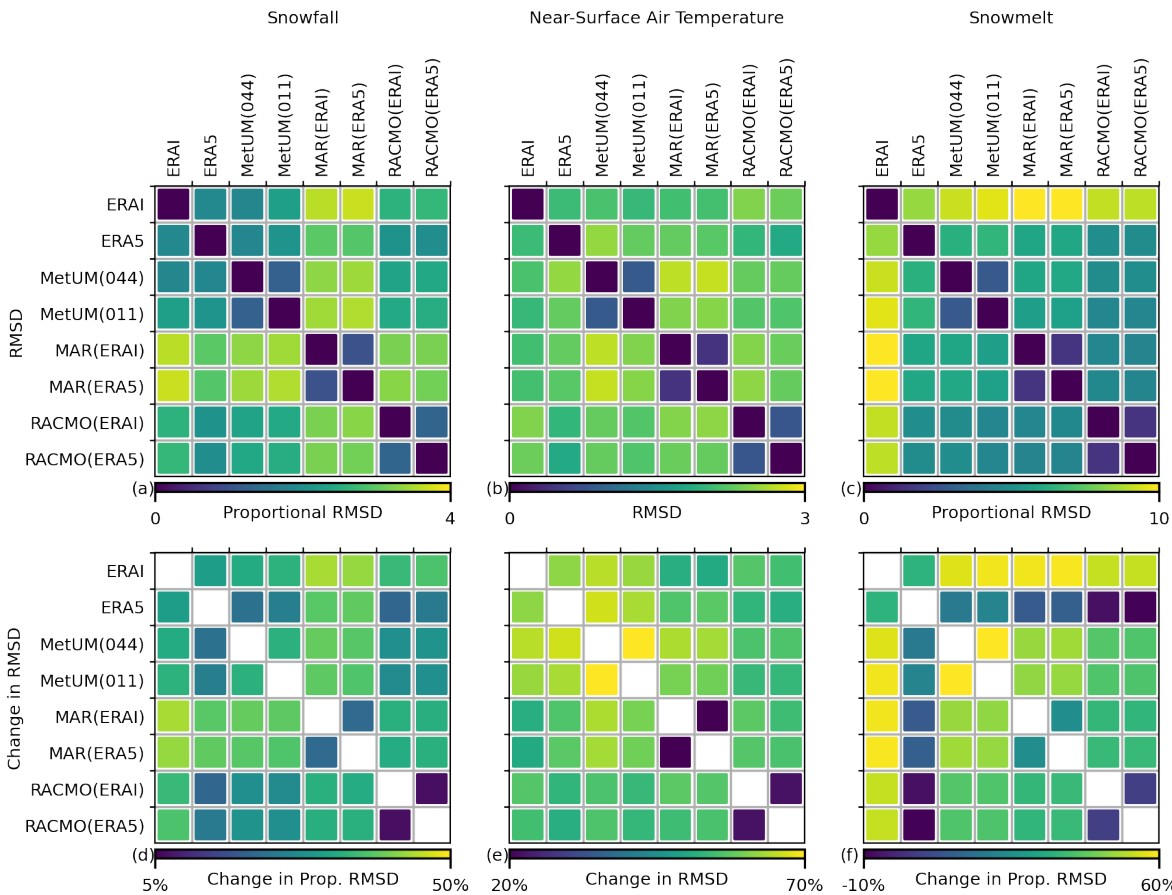

**Figure 7.** The RMSD/proportional RMSD for snowfall (a), near-surface air temperature (b) and melt (c) between models averaged over the ice-sheet. After adjusting the mean and seasonal/residual standard deviations of all outputs to the ensemble average the percentage reduction in RMSD/proportional RMSD is plot for snowfall (d), near-surface air temperature (e) and melt (f).





From Fig. 7 it can be seen the average values for the RMSD/proportional RMSD are significant for all variables, with upper thresholds of 3 K for near-surface air temperature and proportional values of 4 for snowfall and 10 for melt. Values are

comparatively low between simulations of the same RCM but differing resolution/driving data (MetUM(044)-MetUM(011), MAR(ERAI)-MAR(ERA5) and RACMO(ERAI)-RACMO(ERA5)) when compared with differences present between the different RCMs. For melt, ERA-Interim has noticeably higher values of proportional RMSD compared to the other models, while for snowfall and temperature differences are less pronounced but the two simulations from MAR show higher average proportional RMSD for snowfall compared with the other models.

The percentage change in RMSD/proportional RMSD after adjusting for equal means and seasonal/residual standard deviations is significant for all variables, as shown in Fig. 7. Upper thresholds on the percentage reduction are 50%, 70% and 60% for snowfall, near-surface air temperature and melt respectively. For melt, the most significant reductions are for ERA-Interim, while ERA5 shows the least significant reductions with proportional RMSD actually increasing between ERA5 and the RACMO products. Across the variables it can be seen that the percentage reduction in RMSD between the high/low reso-

lution MetUM simulation pairs is of greater magnitude than reductions between the two ERA-Interim/ERA5 driven RACMO pairs and two ERA-Interim/ERA5 driven MAR pairs.

## 5 Discussion

The results presented in this paper show that for all variables studied, when considered across the entire ice sheet, the outputs that came from the same model (MetUM(011/044), MAR(ERAI/ERA5), RACMO(ERAI/ERA5)) exhibit the highest correla-

tions in the time series as well as the smallest systematic differences and RMSDs. This is despite significant differences in resolution between the MetUM runs, which span the highest and lowest resolution RCM simulations made available from the Antarctic-CORDEX project, as well as significant differences in the driving data for the two MAR and RACMO runs. Note that, although ERA5 is an update to ERA-Interim, the results in Table 2 and in section B of the appendix show that the magnitude of systematic differences in the mean and standard deviations between the reanalysis datasets are of similar or greater

magnitude to that of differences between different RCM outputs (Fig. 4, 5 and 6). Updates in the model physics and assimilation techniques used by ERA5 (Hersbach et al., 2020) compared to ERA-Interim are hypothesised to be the primary reason for large-scale differences in snowfall and near-surface air temperature identified between the reanalysis outputs. The particularly significant difference (over an order of magnitude) for ice-sheet-wide melt between ERA-Interim (15.5 GT) and ERA5 (1.1 GT) is hypothesised to be primarily due to an updated surface scheme (HTESSEL) used in ERA5 that allows run-off (Balsamo

et al., 2009).

Results therefore suggest, that differing resolution and driving data are not primary contributors to large-scale spatial variability across the ensemble. Similarity in the spatial and temporal patterns between Antarctic-wide outputs of the same RCM with different driving data agrees with findings from (Agosta et al., 2019), where outputs from MAR are compared with differing reanalysis driving data from ERA-Interim, JRA-55 and MERRA-2. At finer, more localised scales differing resolution





is shown to create significant differences in the mean and seasonal/residual standard deviations for the monthly time series of
each variable, see Fig. B1d, B2d and B3d in the appendix.

The same-model RCM simulations in the ensemble, as well as having identical model physics, parametrisation and tuning,
also share factors such as the domain specification, ice mask applied, digital elevation model and boundary conditions. The
relative contribution of these additional factors is explored in section 5.1 and from this it is argued that the joint influence
of choices in model physics, parametrisation and tuning is the primary factor influencing large-scale variability across the
ensemble. The impact of parameter tuning is discussed in (Gallée and Gorodetskaya, 2008), where it is shown that adjustment
of the parameter for the relative contribution of snow particles compared to ice particles in the radiative scheme used by MAR
has a significant impact on near-surface air temperature as a higher relative contribution of snow particles leads to greater
flux in long-wavelength downwards radiation. In addition to exploring the relative contribution of different factors to the large
spatial scale variability in the ensemble, in section 5.2 specific features of the variability that are mentioned in results section 4
are discussed and the nature of variability for different variables, regions and time-scales is examined.

## 5.1  Contribution to Variability from the choice of: Domain; Ice Mask; DEM and Boundary Conditions

The exact spatial domains differ between the RCM simulations as shown in Fig. 1. However, the spatial domain for all RCM
simulations examined is Antarctic-wide and domain boundaries all exist over the ocean, implying there should be no strong
local forcing at any of the boundaries. The effect of increasing domain size over the ocean on the output of climatology
over the Greenland ice sheet for MAR has previously been studied and found to not significantly impact results over the ice
sheet (Franco et al., 2012). In general, the domain size should be great enough such that the buffer zone, in which boundary
conditions are applied, does not intersect the region of interest, which in this case is the Antarctic ice sheet. It is found that
for the MetUM(044) run the buffer zone intersects some areas of the periphery of the ice sheet, shown clearly in Fig. B1d.
Despite this, it can be seen that effects are localised to the buffer zone boundary and that even for the regions of the ice sheet
that intersect this the relative impact on systematic differences appears less significant than other factors explored. Overall it is
assumed that, for the ensemble of RCM simulations studied, differences in the domains does not have a significant effect on
the model output for surface climatology over the ice sheet.

As well as having differences in the outer domain boundaries, the different models also have slight differences in the specified
boundaries of the ice sheet due to different coordinates and ice masks used. This creates edge effects at the periphery of the
ice sheet, particularly noticeable for melt in for example Fig. 5d at the edges of the Ronne-Filchner and Ross ice shelves. It
is shown in Mottram et al. (2021) and Hansen et al. (2022) that these edge effects, due to inconsistent ice masks, can have a
significant impact on the total estimated SMB over the ice sheet. In this paper, although ice sheet wide totals are computed
(Table 2), the focus is primarily on evaluating variability in the time series at each 12 km grid cell after regridding products to
a common high-resolution grid. Results for spatial maps of differences for melt are masked using an ice-sheet only mask from
Depoorter et al. (2013), which is found to exclude the most significant edge effects from areas where low-resolution models
overestimate the extent of the ice sheet after regridding. The same mask is applied before calculating average correlations
and RMSDs also reducing the impact of edge effects. Results presented and discussed here, particularly regarding large-scale





spatial patterns, are therefore assumed to not be significantly impacted by the different ice masks used in the ensemble of
simulations.

Another important consideration when comparing RCM simulations is how the method of applying boundary conditions
varies across the ensemble. In particular, although all RCMs examined are nudged at the boundaries within buffer zones, MAR
and RACMO also use spectral nudging that constrains the large scale circulation in the interior of domain, while the MetUM
instead uses a re-initialisation procedure. Spectral nudging involves applying the relaxation technique throughout the interior
of the domain to the long wavelength components of the climate model fields (von Storch et al., 2000). This constrains the
large-scale climatology of the RCM output to that of the driving data, while allowing value-added by the RCM in the small-
scale features. The same is aimed to be achieved with the re-initialisation of the MetUM throughout the domain every 12 hours,
as discussed in section 2.4. The fact that systematic differences between the MetUM, MAR and RACMO are all of significant
and comparable magnitude (Fig.4(g,j),5(g,j) and 6(g,j)), despite MAR and RACMO sharing the technique of spectral nudging,
suggests that differences between the specific approach of applying large-scale constraints within the ensemble of RCMs
studied is not one of the main features contributing to variability in the mean and seasonal/residual standard deviations of
snowfall, near-surface air temperature and melt. It is noted however, that in general the MetUM simulations, rather than the
MAR and RACMO simulations, show slightly higher correlation to the reanalysis driving data across the ice sheet for the
monthly time series of snowfall and surface temperature (Fig. 2), indicating that the re-initialisation procedure potentially
constrains the output across the ice sheet more than spectral nudging.

Another feature that the MAR and RACMO simulations share but that differs in the MetUM is having comparatively similar
elevation specified from DEMs. The difference in elevation used by each model relative to the ensemble average is plot in Fig.
C1 in the appendix. The elevation profiles can be split into three main groups: the coarse elevation profile of ERA-Interim (Fig.
C1a); the elevation profile of MAR and RACMO (Fig. C1c,d); the elevation profiles of ERA5 and the MetUM high- and low-
resolution runs (Fig. C1b,e,f). Differences in the elevation profile used for either MAR/RACMO compared with the MetUM
are of much greater magnitude than differences shown between the different resolution MetUM runs. Despite the fact MAR
and RACMO use comparatively similar DEMs, the magnitude of the differences in the mean and seasonal/residual standard
deviations between the models is of similar magnitude as those present between MAR/RACMO and the MetUM, indicating
differences in the DEMs are again not primary contributors to systematic differences in the models output. This is supported
by further results displayed in Fig. C2 where weak linear correlation is found between differences in elevation and differences
in mean near-surface air temperature between a sample of model pairs.

## 5.2   Specific Features of the Variability

Specific features in the variability, identified and mentioned in section 4, are discussed here. In section 4.1 it is mentioned that
for melt there is a clear divide in the average correlation in the residual component of the time series between reanalysis datasets
compared with RCMs (Fig. 2). That is the correlations between different RCMs are greater than between reanalysis datasets
and RCM outputs. This is not the case for snowfall and near-surface air temperature, suggesting the divide in correlation for
melt is primarily due to differences in the sophistication and polar specific tuning of the surface schemes used for the RCM





simulations and the global reanalysis products. It is shown in Hansen et al. (2021) that the subsurface scheme and the handling of layers within the scheme can have a significant impact on melt.

In section 4.1 it is also shown that, particularly for snowfall and melt, the median correlation between the outputs is strongly dependent on the specific region and topography. For melt three regions are highlighted that show low correlation: the Ronne ice shelf; the ice shelves bounding Victoria Land; and the interior of the Amery ice shelf. In the case of the Ronne ice shelf, the low correlation in melt is due to relatively low average melt occurring over the region, so fluctuations away from no melt are small and erratic. Low correlation over ice shelves bounding Victoria Land is expected to be caused by a combination of

their fine scale and the sharply varying topography in the region, making the climatology around them difficult to resolve with the resolution available in the climate models. Finally, the pattern of low correlation around regions such as the interior of the Amery ice shelf is likely the result of atmospheric processes difficult to represent fully in the models, for example: katabatic winds, driven by gravity, flowing from the interior of the ice sheet to the exterior down elevation channels have a significant impact on the climate on the Amery ice shelf, particularly near the grounding line (Lenaerts et al., 2017).

As with for correlation, the systematic differences shown between the outputs in the ensemble vary depending on the region and topography, see section 4.2. This is true at large and small spatial scales and for all variables. An example of a dependency at large scale is in Fig. 4g MAR shows a significant positive difference in the 40-year mean monthly snowfall relative to the other outputs over the majority of the ice sheet and a significant negative difference over the majority of the surrounding ocean. In the case of MAR this is hypothesized to be due to a couple of reasons: MAR is forced at the boundaries by humidity and

needs time to transform this into precipitation; MAR allows precipitation to be advected through the atmospheric layers until reaching the surface. The advection of precipitation in MAR through each atmospheric layer, in comparison to the instantaneous depositing of precipitation by RACMO, leads to increased snowfall towards the interior of the ice sheet, previously identified in (Agosta et al., 2019).

     In section 4.3 the RMSD between each model pair, calculated at each grid cell and then averaged across the ice sheet, is

presented. This metric of average deviation is dependent on the correlation and presence of systematic differences between the outputs. High values of proportional RMSD for melt, shown in Fig. 7 are the result of relatively low correlations between models as well as relatively high systematic differences. It is noted that for melt, despite there being a clear divide in correlations between reanalysis datasets and RCMs (Fig. 2, the RMSD between ERA5 and the RCMs is of comparable magnitude to values between the RCMs. This is due to particularly low values of total melt exhibited from ERA5 (Table 2) and resulting low

magnitude fluctuations of melt. The percentage change in RMSD, after adjusting the mean and seasonal/residual standard deviations of all outputs to the ensemble average, further supports this as it can be seen for melt that ERA5 exhibits the smallest reduction in RMSD after adjustments (Fig. 7). The converse of the above argument is true for ERA-Interim that shows particularly high values to total melt and so particularly significant values of porportional RMSD and of percentage reductions after adjusting the mean and seasonal/residual standard deviations.



## 6 Conclusions

The spatial nature and magnitude of variability present in an ensemble of current, state-of-the-art Antarctic-wide RCM outputs and global reanalysis datasets is examined for snowfall, near-surface air temperature and melt. This is done at a 12 km grid level, rather than across elevation bins, which reveals significant spatial patterns in correlation and systematic differences in the mean and seasonal/residual standard deviation. Time series decomposition is used to split comparisons across an ~inter-annual trend component, a periodic seasonal component and a monthly residual component, which is useful for impact assessments where knowledge of variability in the climate data across different time-scales and climate drivers is needed.

It is found that the RCM outputs and reanalysis datasets show high correlation for the monthly time series of snowfall and surface temperature across the majority of Antarctica and the bounding Southern Ocean. Despite this there exists significant differences, with respect to both magnitude and spatial scale, in the mean and seasonal/residual standard deviations of the time series. In addition, high RMSD between the outputs is found for all variables and is particularly significant for melt with respect to the proportional values relative to annual fluctuations. The primary sources of large-scale, systematic differences between the simulations, for all variables and components, are identified as deriving from differences in: the model dynamical core; the surface scheme; parametrisation and tuning. Differences in driving data, resolution, domains, ice masks, DEMs and boundary conditions are identified as having a secondary contribution. On local, fine spatial scales the relative contribution from different factors is more complex and differences in for example resolution are shown to have a more significant impact.

The variability in snowfall, near-surface air temperature and melt shown is expected to introduce significant uncertainty in estimates of the ice shelf stability with regard to collapse events, which as discussed may have an important contribution to 2100 SLR estimates. It is suggested that the magnitude and scale of systematic differences across the ensemble precludes the direct use and interpretation of individual outputs in impact assessments regarding ice shelf collapse. Results show that removing systematic differences between the ensemble of outputs, significantly reduces the average RMSD. Therefore, as concluded in Mottram et al. (2021), there is an importance on observational campaigns to correct for biases. In addition, further development of RCMs, with particular focus on improvements to the performance of surface schemes over regions of high-melt, is needed to reduce uncertainties around collapse events and 2100 SLR. Finally, it is suggested that further development of sophisticated techniques for bias correction are needed, that are compatible with sparse observations and make use of factors such as the spatial distribution of variability identified in this paper.





*Code and data availability.* The monthly output from all RCM simulations examined in this paper, as well as the processed data used for figures and tables, is available at: https://doi.org/10.5281/zenodo.6367850 (Carter et al., 2022). The code used to import, process and generate the figures/tables is available at: https://doi.org/10.5281/zenodo.6375205 (Carter, 2022). The reanalysis data is available to download through the C3S climate data store (CDS) (ECMWF, 2011; Hersbach et al., 2018). Further outputs from Antarctic-wide RCM simulations are available from the the Antarctic-CORDEX project: https://climate-cryosphere.org/antarctic/.

## Appendix A: STL Decomposition

Figure A1 shows an example of applying STL decomposition to the time series of snowfall, surface temperature and melt for a grid-cell on the Larsen C ice shelf. The decomposition has been applied to each of the 8 model outputs examined in this paper. The trend, seasonal and residual components are shown next to the original time series. Decomposing the time series into these components allows some features to be extracted. For example, in the case of snowfall and surface temperature the models all show high correlation in the inter-annual trend, although there exists a significant systematic difference in the mean between the models. Similarly, for snowfall and surface temperature there is high correlation in the residual term between the models but there is a systematic difference between the models in the standard deviation of that component. In the case of melt, the correlation is more moderate for the trend and residual components, meaning systematic differences are less obvious. The seasonal and residual components in STL decomposition are defined to have approximately zero mean.



**Figure A1.** An example of STL decomposition applied to the monthly time series of snowfall (a,b,c,d), surface temperature (e,f,g,h) and melt (i,j,k,l) for a grid-cell near the grounding line on the Larsen C ice shelf. The original time series for the years 2000-2010 are shown (a, e, i), as are the trend (b, f, j), seasonal (c, g, k) and residual (d, h, l) decompositions. The model is additive meaning the sum of trend, seasonal and residual components returns the original time series. Parameter values are $n_s = 13$ and $n_t = 21$.

 

## Appendix B: Same Core Model Differences



**Figure B1.** The difference for the 1981-2018 time series of snowfall, in the mean (a,d,g,j), the standard deviation of the seasonal component (b,e,h,k) and the standard deviation of the residual component (c,f,i,l) between the following pairs of outputs: ERA-Interim relative to ERA5 (a,b,c); MetUM(044) relative to MetUM(011) (d,e,f); MAR(ERAI) relative to MAR(ERA5) (g,h,i); RACMO(ERAI) relative to RACMO(ERA5) (j,k,l). Differences at each grid cell are expressed as a proportion of average inter-annual variation and so do not have units.



**Figure B2.** The difference for the 1981-2018 time series of near-surface air temperature, in the mean (a,d,g,j), the standard deviation of the seasonal component (b,e,h,k) and the standard deviation of the residual component (c,f,i,l) between the following pairs of outputs: ERA-Interim relative to ERA5 (a,b,c); MetUM(044) relative to MetUM(011) (d,e,f); MAR(ERAI) relative to MAR(ERA5) (g,h,i); RACMO(ERAI) relative to RACMO(ERA5) (j,k,l).







**Figure B3.** The difference for the 1981-2018 time series of melt, in the mean (a,d,g,j), the standard deviation of the seasonal component (b,e,h,k) and the standard deviation of the residual component (c,f,i,l) between the following pairs of outputs: ERA-Interim relative to ERA5 (a,b,c); MetUM(044) relative to MetUM(011) (d,e,f); MAR(ERAI) relative to MAR(ERA5) (g,h,i); RACMO(ERAI) relative to RACMO(ERA5) (j,k,l). Differences at each grid cell are expressed as a proportion of average inter-annual variation and so do not have units.



## Appendix C: DEM Differences

**Figure C1.** The difference between the DEM used by each climate model is plot relative to the ensemble average (a. ERA-Interim, b. ERA5, c. MetUM(044), d. MetUM(011), e. MAR and f. RACMO). The DEMs are regrid onto the MetUM(011) 12.5 $km^2$ grid for comparison. Units are meters of elevation difference.




**Figure C2.** A density scatter plot showing the correlation between the difference in elevation for each model relative to the ensemble and the difference for near-surface temperature in the mean of the time series (a), the standard deviation of the seasonal component (b) and the standard deviation of the residual component (c). The linear Pearson correlation coefficient is given for each plot.





*Author contributions.* J.Carter: Conceptualization, Methodology, Software, Validation, Formal analysis, Writing - Original Draft. A.Leeson: Conceptualization, Writing - Review & Editing, Supervision. A.Orr: Conceptualization, Data Curation, Writing - Review & Editing, Super-
480 vision. C.Kittel: Data Curation, Writing - Review & Editing. J.M. van Wessem: Data Curation, Writing - Review & Editing.

*Competing interests.* The authors declare that they have no conflict of interest.

*Acknowledgements.* J.Carter is supported by the Data Science for the Natural Environment project (EPSRC grant number EP/R01860X/1). J.M. van Wessem was partly funded by the NWO (Netherlands Organisation for Scientific Research) VENI grant VI.Veni.192.083. Compu-tational resources for MAR simulations have been provided by the Consortium des Équipements de Calcul Intensif (CÉCI), funded by the
485 Fonds de la Recherche Scientifique de Belgique (F.R.S. – FNRS) under grant no. 2.5020.11 and the Tier-1 supercomputer (Zenobe) of the Fédération Wallonie Bruxelles infrastructure funded by the Walloon Region under grant agreement no. 1117545. C.Kittel's was supported by by the Fonds de la Recherche Scientifique – FNRS under grant no. T.0002.16 and the H2020 CRiceS. The code for analysis is written in Python 3.8.12 and makes extensive use the following libraries: Iris (Met Office, 2010); NumPy (Harris et al., 2020); Matplotlib (Hunter, 2007).



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
