# Peer review of "Variability in Antarctic Surface Climatology Across Regional Climate Models and Reanalysis Datasets"

_EGUsphere, 2022_

## Author Comment (AC2)

**Author Response to Referee's Comments**

July 6, 2022

Thank you for the review of the manuscript. We are very grateful for your careful and insightful comments, which have contributed to the improvement of the original manuscript. We have worked hard to incorporate the feedback into the revised manuscript and have detailed here our thoughts and any changes made for each comment individually below. We hope you find the response and changes satisfactory.

Main comment

Comment: As surface melt occurs primarily during the summer, any differences in the seasonal cycle require a bit more explanation. For example, in Figure 6, the differences in the seasonal standard deviation for MAR are quite large over East Antarctic ice shelves. What does this mean? Does this mean that seasons are shorter in MAR? It is also worth highlighting immediately that (here in plain language): the only real estimates of surface melt that matter here are from MAR and RACMO. The simpler surface schemes in ERAI/ERA5/MetUM simply don't capture these processes. I note, however, that the authors have provided a thorough explanation of the underlying physics (and similarly did a great job explaining Agosta's work on how the difference in how precipitation is treated in MAR and may lead to more precip in the interior). It is just worth noting these basic differences at the start (potentially within table 1).

Response: It is agreed that for melt systematic differences in the standard deviation of the seasonal cycle could imply differences in the length of the melt-season and could also be specific to different intensities of melt over only the peak summer months. It is likely both are true and that greater values of standard deviation in the seasonal component are expected to represent both higher magnitudes over peak months as well as a slightly prolonged melt season. A sentence has been added on this in the results section, in the paragraph discussing systematic differences shown for melt (Fig.6): "Unlike for snowfall and near-surface temperature, the relative magnitude of differences across the ensemble in the standard deviation of both the seasonal and residual components of the time series are greater than differences present in the mean of the time series. It is noted that for melt, which occurs primarily over just the summer months, greater values of standard deviation in the seasonal component are expected to represent both/either higher magnitudes over peak months and/or a prolonged melt season." .

It is also agreed that it is worth highlighting immediately in the introduction the strong deficiencies in surface schemes of the MetUM and the reanalysis datasets that are expected to lead to non-physically realistic evaluations of melt. To do this the following sentence is added to the introduction, some of which is moved from the model specifications section: "Another feature of particular note in the MetUM simulations examined here is that a 'zero-layer' surface scheme is used, which has been identified as a major deficiency in simulations, compared with the multi-layer schemes included in MAR and RACMO, due to such impacts as that on heat transfer and not representing the insulating properties of the column of snow (Slater et al., 2017; Walters et al., 2019). It is therefore expected that the MetUM, as well as the reanalysis datasets ERA-Interim and ERA5 that both use a single tile to represent snow, will produce much less physically realistic evaluations of melt than MAR and RACMO."

Specific comments

Table 1: Include a column for time-period at which forced at the boundaries. This is mentioned in the text, but is a very important difference.

A column has been added for the time-period at which forcing is applied to the simulations.

*Potentially my misunderstanding, but how does the value and the magnitude of correlation differ from one another?*

Both refer to the same thing, the important difference between the colour scale and size scale used in Fig. 2 is just that the limits of the colour scale are adjusted for each plot, while the limits for the size scale are kept constant to clearly show differences in correlation between the variables. The caption in Fig. 2 has been updated to hopefully make this more clear: *"The colour scale relates to the value of correlation and the scale is adjusted for each plot. The size of each square also relates to the value of correlation, although is kept constant across the figures, going from 0-1, to make comparisons clear between the different variables.".*

*Fig. 3: Use different color scales for (a) and (b) to make explicit that they are different scales (alternatively, just add a note in the caption). This is stated in text, but easy to misread.*

A note has been added in the caption, as with Fig.2: *"The colour scale relates to the value of correlation and the scale/limits are adjusted for each plot."*

*Line 205 :The impact of systematic differences in snowfall/snow melt on estimates. . .*

This sentence has been adjusted to include the correction: *"The impact of systematic differences in snowfall and melt...".*

*Line 208: For near-surface air temperature, differences. . .*

This sentence has been adjusted to include the grammatical correction.

*Line 245: . . . each component of the timeseries and that for temperature and melt . . .*
This sentence has been reworded to: *"It is noted that the relative magnitudes of standard deviations in each component of the time series depends on the variable and that for temperature and melt the seasonal deviation is dominant, while for snowfall both the seasonal and residual deviations have similar magnitudes"*

*Line 258-263: This is partially mentioned in methods, and should perhaps just be moved there*

It is agreed that the reasoning in these lines could be combined with the section in the methods and end of the method section is adjusted to: *"Differences in the monthly mean and standard deviation of the components are calculated over the 37 year 1981-2018 period. For snowfall and melt, differences at each grid-cell are expressed as a proportion of the respective inter-annual deviations, providing some measure of the relative significance of differences at each location. The impact of systematic differences in snowfall and melt on estimates of ice shelf stability depend not only on absolute magnitudes but also on the relative magnitude against a baseline variance, as well as how this influences the ratio in magnitudes between snowfall and melt. The inter-annual, baseline deviation at each grid-cell is approximated as the ensemble average standard deviation in the trend component of the time series. Results presented in spatial maps then show the relative significance of systematic differences and are not simply dominated by the sites with the highest magnitude snowfall/melt. For near-surface air temperature differences for each grid-cell are not expressed as a proportion and instead simply in degrees Kelvin."* .

*Line 264: "significant systematic" implies a more quantitative assumption. Perhaps use "substantial" or "spatially-coherent"*

This sentence has been adjusted to use 'high-magnitude, spatially-coherent systematic...' : *"In Fig. 4, it can be seen that for snowfall there exists high-magnitude, spatially-coherent systematic differences over both the ocean and ice sheet, particularly in the mean of the time series (Fig. 4a,d,g,j), for each model relative to the ensemble average."*

*Line 268: Possibly worth mentioning the length scale in the Antarctic Peninsula as well.*
It's agreed it's worth mentioning the length scales over the Antarctic Peninsula, a sentence is added with reference to the dependency on direction: *"Regions such as the Antarctic Peninsula exhibit direction dependent length scales, with a comparatively large length scale in the latitude direction and a comparatively short length*

scale in the longitude direction."

Line 270: mean of the time series is highly correlated: as "correlated" is a specific quantitative term here, I think that "has a similar spatial signature" might be more accurate.

The sentence has been adjusted as suggested: "It can be seen that for snowfall the difference present in the mean of the time series has a similar spatial signature and sign as the difference in the standard deviation of the seasonal and residual components (e.g. Fig.4g,h,i)."

Line 281: Similarly (to above) I wouldn't say "weak, negative correlation", but rather something like "contrasting spatial patterns" and specify where.

The sentence has been adjusted as suggested and reference given to particular subplots showing this: "While as for Fig.4 there are similar spatial patterns between the mean temperature difference and the seasonal standard deviation difference, the sign of the differences in Fig.5 is in general shown to change, for example over the majority of the ice sheet in Fig. 5d compared with Fig. 5e."

Line 284/ Figure 5: Point to location of "example grid-cell" in the figure

The location of the example grid-cell has been added to the text for context: "In Fig. A1e in the appendix, which gives the temperature profiles from each simulation over an example grid-cell on the Larsen C ice shelf, a colder mean temperature is shown to be the result of similar summer temperatures with more severe winter temperatures."

Line 294: How meaningful is the seasonal standard deviation if the majority of melt happens in summer? (What does this mean physically? Does this imply that the seasons shift?)

The seasonal component captures the component of the time series that repeats year on year. In the case of melt this captures the periodic nature of significant melt over the summer months with very little melt occurring over the rest of the year. The standard deviation of the seasonal component of the time series then represents the average monthly deviation in this component away from the mean. In general an increase in the average deviation in the seasonal component can be thought of as primarily the result of more intense melt occurring over the summer months. It is also true that an increase in for example the melt season duration would result in an increase in the average deviation in the seasonal component. It is hard to disaggregate these two causes and it is likely that they occur together and that if a model shows greater intensity of melt in the summer months it will also have a slightly prolonged melt season.

Line 295: The physical meaning of this metric isn't entirely clear to me: I understand this as a measure of the bias correcting for the effect of season and residual, but it might be helpful to make this explicit and expand a little more
Line 298: The masking for 1mmW Eq was mentioned in Methods, I don't think it needs to be mentioned again.
Line 310: Again, I'm having a hard time parsing what "adjusting for equal means and seasonal/residual standard deviations" means here. I think it's worth explaining a bit more what this metric means physically (e.g. when correcting for the seasonal effect, the mean trend, the residual is a metric of the physics)

In answer to all the above, this paragraph has been adjusted to make it more explicit what each metric means and the information about masking has been moved to a footnote in the caption of Fig. 7. The updated paragraph is as follows: "The RMSD of the monthly time series is evaluated at each grid-cell for each of the 28 unique output pairs of the ensemble. For snowfall and melt, the metric is scaled at each grid-cell by the ensemble average inter-annual standard deviation, described here as the proportional RMSD value. The average is then taken across the ice sheet for each variable and results given in Fig. 7a-c. The average RMSD across the ice sheet provides a measure of the average deviation between the time series of 2 model outputs at each grid cell, while the average proportional RMSD gives a measure of the average relative magnitude of deviations with respect to inter-annual variability. In addition, the percentage reduction in RMSD/proportional RMSD is evaluated after adjusting the mean and seasonal/residual standard deviations at each grid cell for every model output to that of the ensemble average. The average percentage reduction is then taken across the ice sheet for each variable and results given in Fig. 7d-f. This percentage reduction gives a measure of what proportion of the deviations between the time series

are the result of systematic differences in both the mean and seasonal/monthly fluctuations."

Line 321: The large-scale differences in snowfall (ERAI vs ERA5) are attributed here to model physics, but we're also seeing very large differences in the DEMs. Similarly, Met models show quite similar DEMs (Figure C1)

Differences in the DEMs are argued in section 5.1 to not be primary contributors to systematic differences in the models output. This is argued primarily through the fact that large systematic differences exist between MAR and RACMO outputs despite having similar DEMs. A sentence has now been added in the first paragraph of the discussion drawing attention to this: "While EAR-Interim and ERA5 exhibit large differences in their DEMs (Fig. C1), it is argued in section 5.1 that differences in DEM are not primary contributors to systematic differences in the models output."

Line 342-344. This sentence is a little long and could be broken up.

The sentence has now been broken up to: "The impact of parameter tuning is discussed in Gallée and Gorodetskaya, 2008 where it is shown that adjusting the relative contribution of snow particles compared to ice particles in MAR's radiative scheme has a significant impact on near-surface air temperature. A higher relative contribution of snow particles leads to greater flux in long-wavelength downwards radiation."

Line 354: It is found that for the MeUM(044) run, the buffer...
Line 355: to the buffer zone boundary, and that even...

The grammatical corrections have been made.

Line 386 – 396: This paragraph is a little convoluted generally.

It is agreed that this paragraph could be tidied up and the reasoning clarified. The paragraph has now been updated to the following: "The differences between DEMs used across the ensemble are plot in Fig. C1 in the appendix. The elevation profiles can be split into three main groups: the coarse elevation profile of ERA-Interim (Fig. C1a); the elevation profiles of ERA5 and the MetUM high- and low-resolution runs (Fig. C1b-d); and the elevation profiles of MAR and RACMO (Fig. C1e,f). Differences in the DEMs do not mirror the systematic differences shown in section 4.2. For example, while MAR and RACMO share comparatively similar DEMs, the models do not share similar patterns in systematic differences (Fig. 4, 5, 6). This indicates differences in the DEMs are not primary contributors to systematic differences in the models output, which is further supported by results displayed in Fig. C2 where weak linear correlation is found between differences in elevation and differences in mean near-surface air temperature."

Line 418: this seems to be reversed.(positive over ocean, negative over large region of East Antarctica near the Transantarctic mts.

In Fig 4g the blue over the interior and red over the ocean indicate that MAR has greater snowfall over the interior and less over the ocean compared with the ensemble average. The order of the difference being shown a.k.a. (model-ensemble avg) or (ensemble avg - model) is agreed to not be clear and so a note in figure 4,5 and 6 captions is added clarifying this: "The difference to the ensemble average (model-ensemble avg)..."

Line 424: The use of the term "systematic differences" needs to be more clearly differentiated from "correlation" here. I think what you mean is that "this correlation occurs over large portions of the ice sheet" In general, I think this paragraph needs a summary sentence for melt, e.g. that the melt bias is consistent, and wide-spread, even accounting for the seasonal and trend components.

I believe the confusion here is because the paper mentions both spatial and temporal correlations throughout, in this section the temporal correlation is what is being refered to along with systematic differences. In order to make this more clear this has now been explicitly referred to in the text as temporal correlation.

[revised manuscript text omitted]

---

## Author Comment (AC3)

**Author Response to Referee's Comments**

July 7, 2022

Thank you for the review of the manuscript. We are very grateful for your careful and insightful comments, which have contributed to the improvement of the original manuscript. We have worked hard to incorporate the feedback into the revised manuscript and have detailed here our thoughts and any changes made for each comment individually below. We hope you find the response and changes satisfactory.

Main Comment

Comment: "From your plots, it looks to me as though model differences are generally greater over steep, mountainous terrain (e.g. along the coastline and over the Transantarctic mountains). This highlights differences in the representation of orography and meteorological conditions over orography. Previous studies have demonstrated that a major source of such differences is model resolution, e.g. there are significant differences in the reproduction of influential mountains winds between simulations with grid spacings of 12-km and km-scale (e.g. Orr et al., 2014, 10.1002/qj.2296 for katabatic winds in E Antarctica; Heinemann et al., 2021, 10.3390/atmos12121635 for foehn winds over the Antarctic Peninsula). These differences are particularly pertinent for surface mass balance over ice shelves, since the Antarctic coastline is generally found at the foot of the steep slopes of the Antarctic plateau, and/or is in the vicinity of mountainous terrain. So I'm intrigued as to whether in your results you can see larger systematic differences in the vicinity of steep terrain that can be attributed to resolution (using the two MetUM models), and whether these differences are of sufficient magnitude and spatial scale to be pertinent for ice shelf mass balance. Also, as alluded to above, it could be that even the 12 km MetUM simulations are insufficient to reproduce climatically important influences of terrain-induced airflows. You do mention katabatic winds as a potential source of model differences, specifically on the Amery IS, but I wonder if it might be worth commenting further on the influence of orography on model differences and the implications of this."

Response: It is agreed that from Fig.4,5 model differences in snowfall and near-surface air temperature appear generally greater over steep, mountainous terrain and that this is an important feature worth highlighting. In the results we have therefore added:

Regarding snowfall: "The magnitude of the differences shown over the ice sheet appear greater over sharply varying topography, such as the Transantarctic mountain range and the steep coastal slopes of the ice sheet. An exception to this being high magnitude differences also shown in the mean component over the comparatively flat region of the interior of East Antarctica for the MetUM(011) and MAR(ERA5) (Fig.4d,g)."

Regarding near-surface air-temperature: "The magnitude of differences shown over the ice sheet again appear greater over regions of steep topography, particularly for the MetUM(011) and MAR(ERA5) outputs (Fig.5d,g)."

It is also agreed that a major source of these differences in meteorological conditions over orography is resolution and we have added a section to the discussion on this, quoted below. Reference has been made to the papers highlighted by the referee (Orr et al., 2014; Heinemann et al., 2021) that show the sensitivity of important orography-driven atmospheric processes such as foehn and katabatic winds to resolution. The paper by Orr et al., 2021 (https://doi.org/10.1002/qj.4138) is also referenced, which shows foehn wind sensitivity to resolutions at km and sub-km scale over Larsen C. The important influence of foehn and katabatic winds on climatology over ice shelves (typically in the close vicinity of mountainous terrain) is mentioned and reference made to papers in the literature that explore this, including: Bromwich 1989 (https://doi.org/10.1175/1520-0477(1989)070<0738:SAOAKW>2.0.CO;2); Cape et al., 2015 (https://doi.org/10.1002/2015JD023465); Lenaerts et al., 2017 (https://doi.org/10.1038/NCLIMATE3180); Datta et al., 2019 (https://doi.org/10.1029/2018GL080845); Elvidge et al., 2020 (https://doi.org/10.1029/2020JD032463)).

The high-magnitude localised systematic differences over mountainous terrain present in direct comparisons between the high/low resolution MetUM simulations (Fig.B1d, B2d and B3d) are highlighted. Further detail is provided for Fig.B2d where the difference in the mean near-surface air temperature, due to resolution, clearly extends over ice shelves such as the interior of the Amery ice shelf, which is a well-known katabatic wind confluence zone (Parish and Bromwich, 2007). It is suggested, that greater magnitude systematic differences in melt shown in Fig.6 compared to Fig B3(d-f) indicate the importance of different surface schemes over resolution in influencing variability across the particular ensemble of outputs studied. Finally, as mentioned in the referee's comment, we specify that even at 12 km resolution climatically important terrain-induced atmospheric processes, such as foehn/katabatic winds, are likely not being realistically resolved and this is explored further in Orr et al. 2021 where output from the MetUM RCM at 4 km, 1.5 km and 0.5 km during a foehn wind event on the Larsen C ice shelf show no obvious convergence towards observations during the event.

Section added to discussion: "At finer, more localised scales differing resolution is shown to create significant differences in the mean and seasonal/residual standard deviations for the monthly time series of each variable, see Fig. B1, B2 and B3 (d-f) in the appendix that show direct comparisons between the high and low-resolution MetUM simulations. The magnitude of differences in snowfall and near-surface air temperature due to resolution are greatest over regions of sharply varying topography, such as: the Transantarctic mountains; the coastal slopes of the ice sheet; and the Antarctic Peninsula. The representation of atmospheric processes occurring over mountainous regions including foehn winds that occur over the Antarctic Peninsula and katabatic winds occurring over the coastal slopes of East Antarctica are known to be resolution dependent (Orr et al., 2014; Heinemann and Zentek, 2021; Orr et al., 2021). Foehn and katabatic winds have been shown to impact climate over ice shelves, which are often in close vicinity of steep terrain, and are an important driver of surface melt (Bromwich 1989; Cape et al., 2015; Lenaerts et al., 2017; Datta et al., 2019; Elvidge et al., 2020). In Fig. B2d the difference in the mean near-surface air temperature, due to resolution, extends over ice shelves such as the interior of the Amery ice shelf, which is a well-known katabatic wind confluence zone (Parish and Bromwich, 2007). Despite this influence of resolution on the climatology over ice shelves, greater systematic differences in melt shown in Fig. 6 compared with Fig. B3(d-f) indicate the potentially more significant importance of differences in surface schemes across the ensemble of RCMs studied. It is expected that even at 12 km resolution climatically important terrain-induced atmospheric processes, such as foehn/katabatic winds, are not being realistically resolved as is shown in Orr et al. 2021 where output from the MetUM RCM at 4 km, 1.5 km and 0.5 km during a foehn wind event on the Larsen C ice shelf show no obvious convergence towards observations during the event."

Further specific comments

Line 8: "suggested" here is too weak. Suggest instead "Our results imply that..."

This is agreed and line 8 is updated to "Results imply that ... ".

Line 26: "The primary method of ice shelf retreat is through oceanic basal melting". I think this statement requires further qualification. Specifically, adding the word "currently", and something like "with the notable exception of some of the ice shelves on the Antarctic Peninsula (Pritchard et al., 2012)". Recent climate/ice-sheet modelling studies indicate that atmosphere-driven hydrofracture has in the distant past been, and will in the future be, the principal cause of Antarctic ice-shelf collapse (e.g. DeConto et al., 2021, 10.1038/s41586-021-03427-0; Pollard et al., 2015, 10.1016/j.epsl.2014.12.035).

It is agreed further qualification is beneficial here and the paragraph corresponding to line 26 has been updated to specify that this is true for the current climate and when considering the ice sheet as a whole, with notable exceptions being specific ice shelves such as Larsen A and B. In addition, a sentence towards the end of the paragraph is added specifying the critical importance of hydrofracture in distant-past and near-future SLR contributions from Antarctica, as is recommended by the referee.

Updated paragraph: "The primary method of ice shelf retreat, when considered across the entire ice sheet, is currently through oceanic basal melting (Pritchard et al., 2012; Paolo et al., 2015), although notable exceptions are recent and dramatic collapse events, such as the disintegration of the Larsen B ice shelf in 2002, which are linked to anomalous atmospheric conditions through the process of melt-induced hydrofracture (Scambos et al., 2000; van den Broeke, 2005; Bell et al., 2018). Anomalously high near-surface air temperatures (leading to enhanced melt events),

as well as low accumulation (leading to reduced pore space of surface snow), result in greater lateral propagation of melt water into crevasses across the ice shelf, which then deepen due to increased hydrostatic pressure (Kuipers Munneke et al., 2014). This process reduces the structural integrity of the ice shelf and, in addition to fractures created through supraglacial lake filling and drainage, can eventually lead to collapse (Banwell et al., 2013; Kuipers Munneke et al., 2014). Recent ice sheet modelling studies indicate the critical importance of atmosphere-driven hydrofracture events in distant-past SLR variation (Pollard et al., 2015) and near-future 2100-2300 SLR estimates, particularly under high-emission scenarios (DeConto et al., 2021). Comprehensive spatiotemporal estimates of near-surface air temperature over Antarctica, as well as the accumulation of snowfall and quantity of melt water, are thus important for SLR predictions and are typically provided by RCMs (van Wessem et al., 2018; Agosta et al., 2019; Mottram et al., 2021)."

Figure 3c: Why are the interiors of the Ross and Filchner-Ronne ice shelves masked out?

The reasoning for this is mentioned on line 220 in the section of results on correlation and the same reasoning is assumed for subsequent results sections: "grid-cells where the ensemble 40-year average monthly melt is less than 1 millimeter water equivalent per month (mm w.e m-1) are masked as these regions only experience sporadic and insignificant magnitude melt events, essentially equating to numerical noise in the simulations.". Over the interior masked regions of the Ross and Filchner-Ronne ice shelves the ensemble average melt is less than 1 mm w.e. m-1 and so systematic differences across the ensemble are assumed to not be reliable/consistent and not of primary interest in comparisons. A sentence has been added to the captions of Fig. 6 and B3 specifying that grid-cells where the ensemble 40-year average monthly melt is less than 1 mm w.e m-1 are masked.

Figures 4-6: From neither the text nor the figures is it totally clear to me whether the difference is model - ensemble, or ensemble - model. I'd expect it to be the former, and that is indeed my impression from the text. However, "different to ensemble average" implies to me the opposite. Please make this clear, in the text where these figures are first referenced, and in the figure captions.

The text when first referencing the figures has been updated to include a sentence clarifying this: "Differences for each model are then plotted relative to this reduced ensemble average (model-ensemble avg.)". The captions of Fig. 4, 5 and 6 have also been updated for clarification: "The difference to the ensemble average (model-ensemble avg.)...".

Line 446: "The primary sources of large-scale, systematic differences between the simulations, for all variables and components, are identified as deriving from differences in: the model dynamical core; the surface scheme; parametrisation and tuning." In the discussion, the sensitivity of melt to the subsurface scheme is highlighted, and justification is given for the "secondary" importance of the factors listed in the subsequent sentence (driving data, resolution, domains etc.). We may then assume by way of elimination that the model dynamics and physics are the primary sources of systematic differences. I'm not sure though that this reasoning is actually stated, and I think it should be, somewhere in the Discussion section.

This is indeed the reasoning and it is agreed that it should be stated explicitly within the discussion section, therefore the following has been added to the end of section 5.1: "In this section, features including the domain specification, ice mask applied, digital elevation model and boundary conditions applied are argued to not be the primary contributors responsible for the large-scale systematic differences between the ensemble of model outputs. This result, in addition to the previously discussed secondary contributions of resolution and driving data towards large-scale differences, by way of elimination gives that the joint influence of choices in model physics, parametrisation and tuning is the primary factor influencing large-scale systematic differences across the ensemble."

Line 456: "Therefore, as concluded in Mottram et al. (2021), there is an importance on observational campaigns to correct for biases." Do you mean there is demand for new field observations with which to constrain model physics parameterisations? Or for (post-processing) model bias correction? This statement needs expanding on.

It is meant that greater observational spatio-temporal coverage and quality is important for both improving the tuning and updating the model physics and parametrisations, as well as to use and reduce uncertainties in post processing bias correction techniques. The sentence has been expanded upon to clarify this as suggested: "Therefore, as concluded in Mottram et al. (2021), there is an importance on observational campaigns to correct

for systematic differences. Improved coverage and quality of observations will provide greater constraints with which to both tune and update the model physics and parametrisations, as well as to use and reduce uncertainties in post-processing bias correction.".

Line 458: "2100 SLR" A bit specific. Suggest simply "future SLR". The same applies in abstract, line 3.

This has been changed both in the abstract (line 3) and in the conclusions (line 458 - original document) to use "future" rather than "2100".

Technical corrections/suggestions

Line 5: Suggest italicising "Seasonal and trend decomposition using Loess", and also perhaps capitalising the T in trend, to make it clear that this is what STL stands for.

The text has been updated to be in italics and trend has been updated to have a capital T.

Line 168: Suggest italicising "Seasonal and trend decomposition using Loess"

Italics have been added.

Line 201: RMSD: This abbreviation is defined in the abstract, but should be defined when first used in the main text also.

The abbreviation is now also defined in the first instance of RMSD in the text.

Line 220 and other instances: "mmWEqm-1"... I think there should be spaces between the units, so perhaps "mm WEq m-1". Or "mm w.e. m-1" as I've seen this notation used before.

Adjustment of all instances of "mmWEqm-1" in the text have been made to "mm w.e. m-1".

Line 299: Remove repeated number 1 before "mm"

The correction has been made.